# Young Children Benefit from Intensive, Group-Based Pediatric Constraint-Induced Movement Therapy

**DOI:** 10.3390/healthcare12212134

**Published:** 2024-10-26

**Authors:** Katherine S. Ryan-Bloomer

**Affiliations:** 1Department of Occupational Therapy, Rockhurst University, Kansas City, MO 64110, USA; katherine.ryan-bloomer@rockhurst.edu; Tel.: +1-816-501-4381; 2Ability KC, Kansas City, MO 64108, USA

**Keywords:** pediatric constraint-induced movement therapy (pCIMT), group based, preschool-aged children, young children

## Abstract

Background/Objectives: This quasi-experimental study examined the effectiveness of an intensive, group-based pediatric constraint-induced movement therapy (pCIMT) program for young children. Methods: Thirty-five children aged 21 months to 6 years, with unilateral hemiparesis (HP), or weakness on one side of the body from varying etiologies, participated in a 4-week intensive, interprofessional, theme- and group-based pCIMT clinic program in the Midwest, United States. The program ran for 4 weeks with 3 h of therapy per day, 5 days per week with 3 weeks of 24 h casting for the unaffected arm, followed by 1 week of bimanual focus. Outcome measures included the Quality Upper Extremity Skills Test (QUEST), Assisting Hand Assessment (AHA), Canadian Occupational Performance Measure (COPM), and Pediatric Evaluation of Disability Inventory (PEDI). Results: The participants statistically significantly improved the unilateral function of the HP arm in four of five QUEST variables (*p* < 0.009), bimanual coordination as measured by the AHA (*p* < 0.001), and some areas of occupational performance as measured by the COPM (*p* < 0.001) and PEDI (*p* < 0.05). Conclusions: This study revealed the intensive, group-based pCIMT clinic model was effective and feasible to implement with the support from various stakeholders.

## 1. Introduction

Constraint Induced Movement Therapy (CIMT) is one of the most evidence-based interventions for treating children with hemiparesis (HP), or weakness of one side of the body caused by a neurological condition such as cerebral palsy [1,2,3]. Despite the growing research, there are mixed findings regarding the optimal dosage and method of delivery for the young, pediatric population [2,4,5]. Group-based pediatric CIMT (pCIMT) has emerged as an alternative model that combats some of the challenges of signature pCIMT, which additionally provides opportunities for increased social participation in a more cost-effective manner [6,7,8,9,10,11,12]. Group-based pCIMT has been found to be an effective model for producing many benefits, but has typically been implemented with older, school-aged children [9,10,11,13,14,15]. This article aims to describe a real-world intensive, group-based pediatric CIMT (pCIMT program) clinic model for children as young as 21 months up to 6 years of age and evaluate the effectiveness of the program through a pre-post study.

### 1.1. Principles of Pediatric CIMT

Pediatric CIMT (pCIMT) is a neurological rehabilitation intervention designed to improve the function of the hemiparetic upper extremity (UE), or arm, in children [6,16,17]. Pediatric CIMT involves (a) constraint of the less affected upper extremity to facilitate changes in the more impaired UE, (b) use of behavioral techniques of shaping, feedback, and grading, (c) a higher dosage of therapy compared to standard therapy, (d) provision of therapy in a naturalistic setting, and (e) incorporation of a transfer package to carry over gains into daily life ([17], p. 20), [18].

### 1.2. Benefits of pCIMT for Young Children

Following pCIMT intervention, numerous studies found positive gains in the unilateral function of the HP UE, such as increased dissociated movements, grasp and prehension, manual dexterity, muscle activation, quality and quantity of use, and strength [7,11,19,20,21,22]. Pediatric CIMT also produced improvements in bimanual coordination, or the ability of both hands to work together [22,23,24,25]. Additionally, pCIMT intervention led to improved occupational performance, or the ability to perform meaningful and necessary daily life activities [26], achievement of personal goals, and social participation [1,24,27].

Growing evidence reveals that pCIMT may be particularly effective for younger children including infants, toddlers, and preschoolers [3,18,21]. Geerdink et al. [28] found younger children reach maximal learning effects sooner than older children. Sterling et al. [29] purported that younger children produced increased gray matter cortical motor representation following pCIMT. Young children have enhanced neuronal plasticity mechanisms, including overproduction of neurons not yet pruned as well as increased long-term potentiation synaptic production than adolescents or adults [16,30]. Eliasson et al. [4] suggested that pCIMT may be more effective for young children due to their increased brain plasticity and rapid development of the hand during early childhood. Providing pCIMT to younger children requires special consideration of optimal dosage and delivery methods.

### 1.3. Dosage Schedule of pCIMT

The dosage schedule refers to the frequency (number of hours per day), duration (number of days), and length (the number of weeks) that pCIMT intervention is applied [4,18]. The signature pCIMT model involves constraint of the non-affected UE for 90% of waking hours, participation in 6 h of individual therapy per day for 10 days with the inclusion of a transfer package, or home exercises to carry over gains to everyday life [17,31]. Further studies expanded the signature pCIMT model to incorporate 3 h instead of 6 h of therapy per day [22,23,24,25].

Research continues to emerge investigating the optimal intensity of pCIMT [1,3,4,14,22,25,32]. Modified CIMT (MCIMT) models, which involve lower dosage schedules of less than 60 h of total pCIMT intervention spread out over a longer period, generate positive results and may be easier for children and caregivers [28,33]. However, other studies suggest that high, intensive dosage schedules involving more dense delivery of at least 60 h of intensive therapy within a shorter period and longer constraint wear led to better gains following the intervention [14,15,18,32,34]. Ramey et al. [18] attributed statistically greater gains in the high pCIMT intensity with prolonged casting group to more practice time within and outside of therapy. Hybrid models have emerged that incorporate a more intensive focus on bimanual therapy along with elements of pCIMT [2,4,10]. Research suggests the intensity of the dosage and practice may be more important facilitators than the type of training [1,32].

### 1.4. Challenges of Signature pCIMT

Providing pCIMT in children’s homes can create challenges due to the time commitments required of the family [6,33]. Furthermore, compliance of young children wearing the constraint for the required number of hours and days and caregivers facilitating pCIMT activities can serve as additional challenges of signature pCIMT [2,4]. Many pCIMT programs have incorporated strategies to enhance engagement and compliance by creating child-centered themes and group-based delivery models, such as summer camps [2,4,33,35,36]. In a study performed by Weerakkody et al. [35], stakeholders providing or supporting the provision of adult CIMT mentioned several challenges for successful implementation of CIMT, including therapist training, additional staffing to cover caseloads, space, and resources. Many of these same inhibiting factors affect pCIMT delivery. The cost of pCIMT, reimbursement from third parties for services rendered outside of healthcare facilities, and the education routines of children pose additional challenges to the implementation of signature pCIMT [4,15]. Group-based pCIMT models offer an alternative method of delivery to combat the challenges of signature pCIMT [4,6,15].

### 1.5. Benefits of Group-Based pCIMT

Group-based pCIMT involves treating multiple children with hemiparesis at once instead of individually allowing more children to access this intensive intervention than signature models [6]. Ratios of the therapist to the child can vary from 1:1 to a small group of children [4,6,12]. Usually, an occupational therapist (OT) or physical therapist (PT) directs group-based pCIMT programs, but other allied health professionals and students can receive training and support the implementation of pCIMT, which reduces costs [7,8,9,10,11,12].

Group-based pCIMT programs can occur within clinics or community-based facilities, which allow billing through third-party reimbursement systems [12,15,35]. Many group-based pCIMT programs are theme based, which can enhance the sense of “fun” and motivation for the participants [14,28,36,37]. The clinic setting, when disguised as an adventure camp or a pirate’s cove, can simulate a more naturalistic preschool or summer camp environment [6,23,28,37]. Offering high-density summer pCIMT camps allows school-age children to participate in intensive therapy without interrupting their school schedule [15].

Peer competition and social participation serve as additional benefits of group-based pCIMT [38]. The children offer modeling for how to approach new motor movements and can motivate peers to problem-solve through challenging activities [36,37,38]. Parents and children reported high levels of satisfaction with gains made from group-based pCIMT and mentioned that the improvements made outweighed the challenges of transportation and disruption of family life [36,37]. Some families even expressed a desire for the pCIMT program to be longer and offered earlier in childhood [36].

### 1.6. Problem, Purpose, and Hypotheses

There are many benefits to group-based pCIMT delivery; however, the literature reveals much variation with regard to the intensity of dosage and involvement of bimanual therapy within group-based pCIMT programs [7,8,9,10,11,12,14,15,24,37]. Participants in most of the group-based pCIMT studies were school-aged or adolescents and the age ranges of the groups were wide. Few studies have evaluated an intensive, group-based model with younger, preschool-aged children [7,12,19,23,37]. The purpose of this study is to describe the methodology of a real-world intensive, group-based pCIMT program offered in a clinic for children ages 21 months to 6 years of age and to report on results from a pre-post study measuring the program’s effectiveness. The PI hypothesized that following participation in intensive, group-based pCIMT, children would display statistically significant improvements in (1) unilateral function, (2) bimanual coordination, and (3) occupational performance.

## 2. Materials and Methods

### 2.1. Design, Participants and Setting

This study employed a quasi-experimental mixed design [39]. Independent variables included a repeated measures variable of time (pre- and post-intervention) and a between variable of program (2–3-year-old program or 4–6-year-old program). This study examined 16 dependent variables across the unilateral function, bimanual coordination, and occupational performance outcome measures’ subtests and total scores. The study took place in the pediatric unit of a medical rehabilitation facility in an urban, Midwest city in the United States. The camp rooms were decorated with child-friendly decorations to align with the program’s yearly theme.

Inclusion criteria required children to be between the ages of 20 months through 6 years, have a diagnosis of unilateral hemiparesis (HP), be authorized to attend the program, have a Manual Ability Classification Scale (MACS) [40] or Mini-MACS level of 1–3, and be able to follow simple commands. Children were excluded if they were in Child Protective Service custody, if they received additional OT during the program, if they missed more than 3 days, if they did not adhere to the casting protocol, and if they were not participating in the intensive, group-based CIMT program for the first time. Children were recruited using convenience sampling. The case manager sent program brochures around the region and to current families at the facility that met the criteria. Three to six spots were offered per year for both the 2-to-3-year-old and the 4–6-year-old programs, respectively.

Forty children were recruited to participate in the study and 35 children completed the intensive, group-based pCIMT program. Of the five children who did not complete the study, one child missed more than 3 days of the program, one did not complete discharge testing, two children’s families did not follow the required casting protocol, and one child was unable to complete the program due to later insurance denial. The mean age of the participants was 37.5 months (SD = 15.81) with a median age of 33 months. Eighty percent of the sample presented with congenital hemiplegic cerebral palsy, while 20% of the sample sustained hemiplegia from acquired conditions. Refer to Table 1 for demographic information of the participants.

Over the eight years the study ran from 2016 to 2023, seven 2–3-year-old pCIMT camps occurred one time per year each summer with 3–6 children attending the camp per year. During 2020, the 2–3-year-old program was suspended due to the COVID-19 pandemic. The 4–6-year-old program ran one time per year for eight summers with three to six children attending each year. Sixty-eight percent of the sample in this study (*n* = 24) participated in the 2–3-year-old program, whereas 32% of the sample (*n* = 11) participated in the 4–6-year-old program. Numerous children came back for multiple summers to participate in the intensive, group-based CIMT programs, but this study focused on analysis of the children’s first experience in the program. Another study [41] investigated the effect of repeated episodes of the program.

### 2.2. Measures

#### 2.2.1. Quality of Upper Extremity Skills Test (QUEST)

The QUEST [42] is a standardized, norm-referenced performance-based assessment of UE unilateral function for children with cerebral palsy ages 18 months to 8 years, which measures dissociated movements, grasp, weight-bearing, and protective extension of both UEs. The QUEST has interrater reliability for subtests and totals ranging from an ICC = 0.67–0.92, intra-rater reliability ranging from an ICC = 0.88–0.96, and high internal consistency with a Cronbach’s alpha of 0.97 [43].

#### 2.2.2. Assisting Hand Assessment (AHA)

The AHA Kids version 4.4 [44,45] is a standardized criterion-referenced test of bimanual coordination for children with cerebral palsy and obstetric brachial plexus palsy ages 18 months to 12 years. The AHA is a play-based assessment, where a play conductor presents toys and a certified examiner scores the video recording for 22 items following administration. Scores are totaled for a sum score ranging from 22 to 88 and scores are converted to an interval, a scaled score ranging from 0 to 100. The AHA has an internal consistency level of 0.97, inter-rater reliability of ICC = 0.98, intra-rater reliability of ICC = 0.99, and test–retest reliability of ICC = 0.98–0.99. A gain of four logits on the AHA scale indicates clinically significant improvements [46].

#### 2.2.3. Canadian Occupational Performance Measure (COPM)

The COPM [47] is a self-report assessment of occupational performance and satisfaction with the performance of meaningful, individualized goals. Caregivers of clients under the age of eight identify the five most important areas of occupational difficulty within the domains of self-care, productivity, and leisure and rate them on a ten-point scale for performance and satisfaction. The COPM demonstrated consistent test and retest reliability (r = 0.92–0.93) as well as strong validity. A gain of two or more points on the COPM following intervention indicates a clinically significant gain [47].

#### 2.2.4. Pediatric Evaluation of Disability Inventory (PEDI)

The PEDI [48] is a norm- and criterion-referenced assessment that evaluates functional capabilities, caregiver assistance, and modification levels of children aged 6 months to 7 years 6 months of age within the domains of self-care, mobility, and social function. The PEDI displays good psychometric properties with interrater reliability ranging from ICC = 0.95–0.99 for all domains and intra-rater reliability of ICC ≥ 0.99 [49]. The Mobility domain was not analyzed for this study.

### 2.3. Procedure and Intervention

The Rockhurst University IRB committee granted approval for the study (2015-15). The study was retrospectively registered through Clinicaltrials.gov on 19 March 2024 (ID: NCT06330831). The study ran from 2016 to 2023. The PI received CIMT training through AOTA and AHA certification. Two occupational therapists served as clinical investigators (CIs), who received training through the University of Alabama Birmingham CIMT training program, and ran the 4–6-year-old programs. The intensive, group-based pCIMT program was modeled after the Kennedy Krieger Institute pCIMT program following consultation with the OTs [22]. The PI trained student investigators on the administration and scoring of the QUEST, COPM, and PEDI and they achieved good inter-rater reliability (ICC > 0.90) after individually scoring five QUEST videos. The PI scored all AHA pre- and post-assessments and ran the 2–3-year-old program.

The intensive group-based pCIMT programs ran for 3 h per day Monday–Friday over 4 weeks during the summer. The CIs applied casts to the non-affected UEs from mid-humerus to distal fingertips with 90 degrees of elbow flexion, which prevented grasping but availed shoulder adduction and forearm stabilization. The participants wore the cast for at least 90% of a 24 h period for 21 days. The fourth week focused on bimanual skills without the casts. Children attended either the 2–3-year-old or 4–6-year-old program, with younger children attending during the mornings to allow for afternoon naps. The PI and CIs created theme-based lesson plans grounded in pCIMT principles used by both age groups (Table 2). Both age group programs operated similarly with only minor modifications made to the younger group, such as shortening the length of time spent reading stories, using the same welcome song every morning to facilitate a sense of routine, and modifying activities that required higher cognitive or academic skills to be more developmentally appropriate. The OT (PI or CI) was present every hour of the program but cotreated with PT, speech language pathology (SLP), music therapy, art therapy, or adaptive martial arts. The PI and CIs trained pre-healthcare student volunteers to be intervention assistants, which availed for a ratio of 1:2 or 1:1 of interventionists to children. The OTs debriefed with caregivers at the end of each day, provided small weekend home exercise program (HEP) activities, and sent a detailed, individualized HEP following discharge. The OTs did not formally track HEP implementation or compliance.

### 2.4. Data Analysis

The PI analyzed the data with Statistical Program for the Social Sciences (SPSS) version 29. The PI evaluated if there were differences in outcome measures between the two pCIMT programs, and thus considered program age-group as a between-subjects variable. The PI ran mixed factorial 2 × 2 ANOVAs for each dependent variable due to having two independent variables of time (repeated measures) and program (between subjects). Since there were 16 dependent variables (five unimanual QUEST variables, one bimanual coordination AHA variable, and ten occupational performance COPM and PEDI variables), the PI evaluated pairwise comparisons using a Bonferroni adjustment to reduce chances of committing a Type 1 error [50]. Levene’s Test of Equality was run to analyze if there were group differences between age groups. Partial Eta Square (η^2^) measure of effect size was selected for analysis for the mixed factorial ANOVAs A η^2^ = 0.01 indicates a small effect size, η^2^ = 0.06 indicates a medium effect size, and a η^2^ > 0.14 indicates a large effect size. Alpha (α) level of *p* < 0.05 was selected to identify statistical significance. The PI additionally ran frequency analyses and descriptive statistics to analyze demographic data.

## 3. Results

### 3.1. Unilateral Function Results

Levene’s test of Equality revealed no statistically significant variance between age groups of any of the QUEST variables. The mixed factorial ANOVA tests revealed no statistically significant main effects for the age group between the 2–3-year-old and 4–6-year-old programs. The mixed factorial ANOVA tests revealed a statistically significant main effect for time with four of the five QUEST variables displaying statistically significant improvements from pre- to post-intervention as seen in Table 3. Though the QUEST Grasps subtest total scores improved from pre- to post-intervention, the scores did not statistically significantly improve. The mixed factorial ANOVA tests indicated there was only one statistically significant interaction effect for age group vs. time occurring in the QUEST Weightbearing subtest, *F*(1, 35) = 4.3 and *p* = 0.045, as depicted in Figure 1. Children in the 2–3-year-old group displayed lower scores at pre-intervention (*x* = 48.64, SD = 30.7) compared to the 4–6-year-old group (*x* = 52.32, SD = 28.73); however, they accelerated at a higher rate by post-intervention (*x* = 67.31, SD = 21.78) than their 4–6-year-old counterparts (*x* = 54.91, SD = 28.46).

### 3.2. Bimanual Coordination Results

Levene’s test revealed no statistically significant variance between the age groups. The mixed factorial ANOVA test indicated no statistically significant main effect for the age group on the AHA. A statistically significant effect for time was detected as the entire sample displayed statistically significant improvements from pre- to post-intervention for the AHA scores, as depicted in Table 3. A statistically significant interaction effect occurred, *F*(1, 35) = 4.46 and *p* = 0.039. The children illustrated the same trend as with the QUEST Weightbearing subtest where the 2–3-year-old group displayed a lower pre-intervention score (*x* = 33.5, SD = 19.45) than the 4–6-year-old group (*x* = 34.36, SD = 19.74) but surpassed the 4–6-year-old group post-intervention score (*x* = 42.54, SD = 21.24) with a score of (*x =* 47.5, SD = 22.48). The pairwise comparisons revealed both age groups statistically significantly improved in bimanual coordination from pre-post intervention with both groups displaying differences at the adjusted *p* < 0.001 level. Figure 2 represents the AHA score obtainment for both age groups at pre- and post-intervention.

### 3.3. Occupational Performance Results

#### 3.3.1. COPM Results

Levene’s test revealed no statistically significant variance between the age groups on the COPM. The mixed factorial ANOVA revealed a non-statistically significant main effect for groups. There was a statistically significant main effect on time for both performance and satisfaction regardless of age group as depicted in Table 4. There was no statistically significant interaction effect for time and group.

#### 3.3.2. PEDI Results

Levene’s test revealed no statistically significant variance between the age groups on the PEDI variables. The mixed factorial ANOVA revealed statistically significant differences between program age groups for both the PEDI Selfcare Functional Skills Normative sores, *F*(1, 35) = 30.23, *p* < 0.001, and η^2^ = 0.478, and PEDI Selfcare Caregiver Assistance Normative scores, *F*(1, 35) = 24.66, *p* < 0.001, and η^2^ = 0.428, as depicted in Table 5. However, this finding is not surprising as the normative scores are based on age and one would expect the scores of the two age groups to be different. The 2–3-year-old group displayed higher PEDI Selfcare Normative scores regardless of time (*x* = 38.29) than the 4–6-year-old group (*x =* 18.24). Similarly, the 2–3-year-old group achieved higher PEDI Selfcare Caregiver Assistance scores regardless of time (*x* = 39.45) than the 4–6-year-old group (*x* = 20.1). These findings indicate that the younger groups’ scores were closer to those of the typical population and thus they achieved higher normative scores than the 4–6-year-old group. As children with disabilities age, the gap in ability level may widen more from the typical population; thus, the children in the older group may have achieved lower normative function, regardless of the effect of the intervention.

The mixed factorial ANOVA tests and pairwise comparisons revealed a statistically significant main effect for time, regardless of age group, for the PEDI Selfcare Normative scores and PEDI selfcare caregiver assistance scaled scores as seen in Table 4. No statistically significant interaction effects were detected.

For PEDI Social Function variables, Levene’s test revealed no statistically significant variance between age groups. The mixed factorial ANOVA test produced a statistically significant main effect for the program age group, regardless of time, for PEDI Social Function Normative scores, *F*(1, 35) = 17.1, *p* < 0.001, η^2^ = 0.341 and for PEDI Social Function Caregiver Assistance Normative scores, *F*(1, 35) = 9.07, *p* = 0.005, η^2^ = 0.216. The means for the 2–3-year-old group were statistically significantly higher for PEDI Social Function Normative scores (*x* = 46.15) than the 4–6-year-old group scores (*x* = 27.18). The PEDI Social Function Caregiver Assistance Normative scores illustrated a similar trend with the 2–3-year-old group displaying statistically significantly higher scores (*x* = 47.65) than the 4–6-year-old group (*x* = 32.44), which are displayed in Table 5. These findings relate to the PEDI Selfcare Normative scores where the younger group’s scores were closer to typically developing peers’ scores and thus produced higher normative scores than the 4–6-year old group. The mixed factorial ANOVA tests also revealed a statistically significant main effect for time, regardless of age group, for Social Function Normative scores, as depicted in Table 4. This finding suggests that the intensive, group-based pCIMT program fostered statistically significant improvements in social function. There was no statistically significant interaction effect for the PEDI Social Function variables.

All PEDI social function scores were higher at post-intervention compared to pre-intervention when both age groups were combined, with the statistically significant effects for time depicted in Table 4. For the 2–3-year-old program, all scores improved from pre-post interventions with the exceptions of Social Function Caregiver Assistance Normative scores and Social Function Caregiver Assistance Scaled scores, which remained similar to pre-intervention levels. The 4–6-year-old scores similarly increased from the pre- and post-intervention for most occupational variables except PEDI Selfcare Functional Skills Scaled scores and PEDI Social Function Functional Skills scaled scores, which remained similar at pre- and post-intervention. It is interesting to note that the mean normative scores of the 2–3-year-old participants were higher at both pre- and post-intervention than the 4–6-year-old participants, whereas the scaled scores were higher for the 4–6-year-old participants at both data collection points. PEDI Scaled scores are criterion-referenced and determined by the number of skills within a continuum the child is able to do, and the level of caregiver assistance needed for particular selfcare and social function areas. Thus, the older children may have accomplished more of these skills or needed less assistance due to their age and developmental status than the younger children.

Only one adverse event occurred where a child pulled her unaffected UE out of the permanent cast, which resulted in skin lacerations. Medical staff treated the skin lacerations, placed a long, Pedi-Wrap over the bandages with a soft mitt over the hand of the non-affected UE and the child continued the final two days of constraint wear and remaining bimanual week.

## 4. Discussion

This study sought to describe the methodology of a real-world, non-externally funded clinic-based intensive, group-based pCIMT program and to measure the effectiveness of the program for children between 21 and 6 years of age. The PI accepted hypothesis one (unilateral function) due to statistically significant improvements from pre- and post-intervention on four of the five QUEST variables. The results reveal that the program was effective in improving numerous outcome areas for the children in this sample. These results corroborate other group-based pCIMT studies that found statistically significant increases in unilateral function on the QUEST [7,11,19,20]. The OTs noted more prehension skill ability during the sessions and parents reported improvements in hand function for many of the participants, such as the way the children attempted to grasp, release, and manipulate objects with the HP hand. However, these qualitative improvements did not translate to statistically improved Grasps subtest scores differing from other studies [7,11,19,20]. Sakzewski et al. [34] found that children who participated in group-based pCIMT improved more on gross motor skills of reach, target accuracy and transfer skills compared to those receiving bimanual therapy who improved more on manipulation, grasp, and release. The results of the current study support the findings of Sakzewski et al., [34] revealing that the children in this sample improved in the gross unilateral function as seen in improvements in dissociated movements, weightbearing, protective extension, and total scores. The Weightbearing subtest statistically significantly improved more dramatically for the younger age group than the older group, which was an interesting finding. Few studies have specifically compared 2–3-year-olds to 4–6-year-olds participating in the same program, and thus drawing conclusions for the discrepancy in the rate of skills and the magnitude of achievements made within this domain is limited. Both programs frequently incorporated developmental positions such as crawling, creeping, and transitioning between movements, which may have led to an improved ability to bear weight through bilateral upper extremities.

The PI accepted hypothesis two (bimanual coordination) due to statistically significant improvements from pre- and post-intervention on the AHA. The positive changes made within bimanual coordination correlate well with findings of other pCIMT group-based intervention studies that utilized the AHA as an outcome measure [7,9,10,14,20,23,24,37]. Metzler [9] found that younger age was associated with AHA gains; however, this was influenced by baseline AHA scores, as children who had the lowest baseline AHA scores were noted to make the most gains following pCIMT. Within this study, the younger age group displayed a lower initial AHA mean score yet surpassed the older group post intervention. Geerdink et al. [28] suggested that older children may have a longer learning curve than younger children and may not show gains as quickly as younger children which may explain why the 2–3-year-old group made more statistically significant improvements than the 4–6-year-old group on AHA skills. Catering activities based on the children’s bimanual level from pre-AHA scores, providing visual cueing on the HP hand with stamps or stickers, and encouraging the integration of the HP UE during the bimanual week may have led to statistically significant bimanual coordination scores. Lee et al. [8] recommended that group-based pCIMT programs have at least one occupational therapist be AHA certified to support goal setting for pCIMT participants. One advantage of the PI scoring the children’s pre-intervention AHA is that the therapist had a better understanding of what specific bimanual skills are on the AHA hierarchy of hand skills to target within an intervention. Though this could be viewed as a bias that led to higher scoring by the PI on the post-AHA for the 2–3-year-old group, for which she implemented intervention, knowledge of pre-AHA skills by the interventionist can also be viewed as a strength of the study. The PI, who was AHA certified, was more keenly aware of the skills to target during the program versus the clinical investigator who ran the 4–6-year-old program, who was less familiar with the AHA and thus may not have targeted the specific AHA bimanual skills as heavily.

The PI partially accepted hypothesis three (occupational performance) due to statistically significant increases in the COPM areas and three of the eight scores of the PEDI. Parents and caregivers noted improvements in both performance and satisfaction of meaningful occupations as evidenced by statistically significant improvements in the COPM which correspond with results of other group-based pCIMT studies [14,23,24,28]. PEDI Self-care functional scores statistically significantly improved aligning with the results of Cohen-Holzer et al. [24]. The statistically significant increase in selfcare normative scores and in the level of caregiver assistance provided measured by caregiver assistance scaled scores from pre-post intervention suggest that the improved hand function may have translated into greater independence with selfcare tasks by the children by the end of the pCIMT intervention, which correlates with other studies [11,35,38].

Few studies have analyzed social function as an outcome measure of pCIMT intervention [11,27]. The implementers of the program selected to assess social function as an outcome measure due to the group-based model of this intervention. By having children participate in pCIMT training alongside other children in a group-based model, there are additional benefits available, such as improved social participation with peers, peer play, and joint attention which are important skills for more successfully engaging in childhood occupations across a variety of home, community, and school settings [10,26]. The statistically significant improvements in the participants’ normative social function scores from pre- and post-intervention correspond with statistically significant gains in social function, following pCIMT intervention reported by other studies [11,27]. These findings reveal that the intensive, group-based pCIMT program may have contributed to children performing more social function skills closer to the level of their peers.

### 4.1. Limitations

This study lacked a control group and randomization. Though the sample displayed good diversity for gender, affected side of HP, cause of HP, and onset type, the majority of the sample was White/Caucasian children and were from the Midwest, United States, which increased sampling bias. There were more children within the 2–3-year-old program than the 4–6-year-old program which may have increased sampling and statistical bias. Additionally, this study experienced historical bias and occurred during the COVID-19 pandemic which affected recruitment. Several raters scored the assessments and the PI scored assessments, ran the 2–3-year-old program, and performed statistical analysis which may have led to measurement bias. The pCIMT program ran for eight years and used different lesson plans implemented by three different OTs in two different age groups across the years which may have elevated intervention bias.

The PI implemented strategies to reduce the limitations, including strict training and inter-rater reliability procedures for raters, pCIMT and casting protocols, use of the same lesson plan structure throughout the years, detailed documentation notes, partial blinding procedures, where the raters scored post-assessments without referencing the pre-assessment scores.

### 4.2. Implications for Future Research

Future studies should focus on measuring the efficacy of group-based pCIMT for various pediatric age groups. Exploratory, predictive studies could examine if certain factors lead to successful outcomes following group-based, intensive pCIMT intervention. Thirteen of the children in this study returned for a consecutive dosage of intensive group-based pCIMT the following summer. Ryan-Bloomer [41] analyzed the effects of long-term outcomes of the pCIMT program as well as the effect of a repeated, consecutive episode of intensive, group-based pCIMT and has submitted the article for publication. Future research could also investigate the responsivity of other assessments that may be less time-consuming to administer and score but produce comparable results to the gold-standard pCIMT assessments.

### 4.3. Implications for Clinical Practice

This study reveals that young children were able to participate in and benefit from an intensive, group-based pCIMT program. This study adds to the evidence base for group-based pCIMT studies for younger children under the age of four years [7,12,20,23,28,37]. Since studies suggest that young children have increased neural plasticity, quicker learning curves, and may respond well to intensive UE rehabilitation interventions, implementation of a high dosage schedule at a younger age may maximize results [18,28,29,30]. The intensive, group-based CIMT program produced similar statistically and clinically significant gains to other group-based programs and signature models [22,23,24,25]. Many of the group-based programs have implemented a shorter duration, decreased frequency of constraint wear, or have not been implemented with children as young as 21 months of age. This study adds a unique perspective in that this group-based program incorporated an intensive model with 60 h of pCIMT therapy over 4 weeks, 24 h casting for 21 days, and primarily served a younger population of 21 months up to 6 years of age, with 77% of this sample being under the age of 4 years.

This intensive, theme and group-based pCIMT program was feasible for the children, families, and the rehabilitation center. Nearly all the young children tolerated prolonged casting and parents reported satisfaction with their children’s improvements during the program corresponding to the results of other studies [18,21]. Across the years that the program was implemented, only one adverse event occurred with a child removing her permanent cast. Though the children were young, they quickly adjusted to the cast, which correlates to the findings of Ramey et al. [18]. The OTs helped to make this transition easier for the children by providing caregivers with a tip sheet with suggestions for providing additional support to reduce frustration. Activities included dressing the child in looser fitting clothing, providing the child with cups with a large handle that could more easily be grasped with the non-affected hand, and providing more physical support as needed. The permanent cast made compliance easier because the parents did not have to take off and reapply for the cast. The 90 degrees of elbow flexion of the casting position allowed the children to still carry objects with both arms and weight bear through their forearms, which additionally made the prolonged casting during the constraint phase more manageable for the young children.

The 3 h morning session for the 2–3-year-old children allowed for naps if needed. The theme-based element of the program availed the opportunity to practice developmentally appropriate social skills and prepare the children for future educational occupations. The social aspect of the program facilitated universality, friendly competition, modeling opportunities, cooperation, and friendship which supports the findings of other pCIMT group-based studies [11,12,36,38]. Through circle time, taking turns, reading stories, and participating in stations, the children participated in educationally relevant occupations that simulated a naturalistic childhood context, which is an essential pCIMT principle [17,18].

Though some studies suggest home-based MCIMT programs where parents implement the pCIMT intervention may be easier on families [21,33], others suggest families may prefer the density of an intensive program within a shorter period provided by therapists [15,32]. Families often recruited relatives or friends to help with daily transportation and therapists relayed home programming through phone if more convenient for working parents. Some of the families in this study did not have access to pCIMT programming within their regions. These families stayed with relatives or at the nearby Ronald McDonald House, a home for families of children living outside of the region receiving hospital or rehabilitation services. These families anecdotally mentioned the inconvenience and cost associated with the intensive program were worth the sacrifice due to the immense gains made in a much shorter amount of time than they had seen during their standard therapy care. This summer intensive, group-based CIMT program offered families an intensive burst of therapy they may not otherwise have received with minimal disruption to educational routines for school-age children.

The program was able to be implemented at the rehabilitation center with support from the administration to provide coverage for the CI’s normal caseloads so the CIs could run the pCIMT programs and develop lesson plans along with funding for pCIMT training; with the collaboration of a university OT program to provide student assistance for evaluations and statistical analysis; with numerous volunteers to serve as intervention assistants; and with parent dedication to bring the children and implement home programming, which parallels the supports needed for successful CIMT implementation reported by Weerakkody et al. [35]. The PI and CIs received advanced training and AHA certification, which increased treatment fidelity and the use of responsive, evidence-based gold-standard pCIMT outcome instruments to measure the effectiveness of the pCIMT intervention, which are essential elements to quality delivery of pCIMT [4]. The OTs billed daily sessions as multidisciplinary therapy group or therapy group charges through third-party reimbursement entities. The intervention assistants availed the OTs to provide a 2:1 or 1:1 ratio of children to an interventionist, which provided more optimal support for the children while lessening the cost burden of multiple therapists providing care [8,12].

## 5. Conclusions

Children in this study made statistically significant improvements in the unilateral function of the HP UE, bimanual coordination, social function, and occupational performance, following participation in an intensive, group-based pCIMT program. This study adds to the pCIMT evidence base as it is one of the few studies to implement a group-based model with prolonged casting, 60 h of therapy within 4 weeks for young children, some below the age of two. The theme and group-based model offered opportunities for social participation, motivation, and preparation for educational occupations in a simulated naturalistic setting while lessening the cost burden for the rehabilitation facility. This study added ecological validity by illustrating how a real-world, non-externally funded, intensive, group-based interprofessional pCIMT model can be implemented in clinics or hospitals with support from administration, dedication of families, and commitment from therapists.

## Figures and Tables

**Figure 1 healthcare-12-02134-f001:**
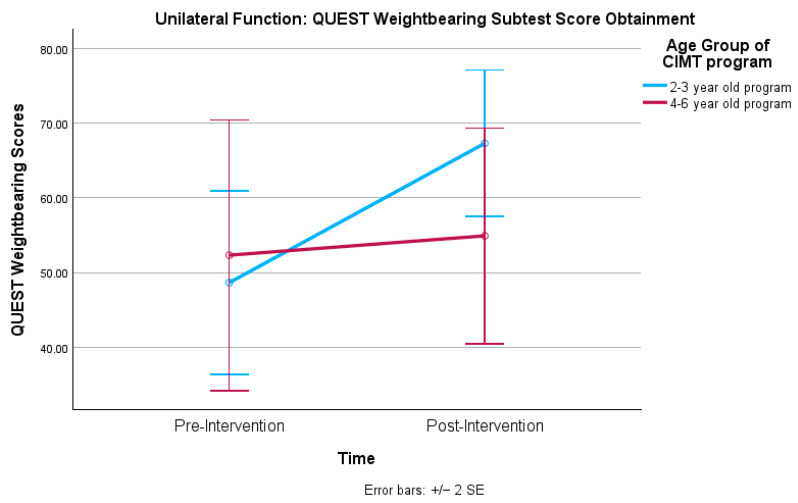
Unilateral function: QUEST weightbearing score obtainment.

**Figure 2 healthcare-12-02134-f002:**
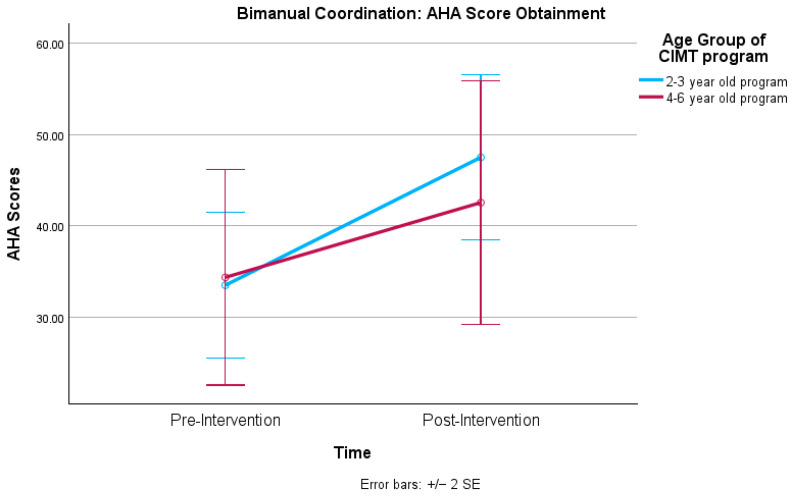
Bimanual coordination: AHA score obtainment.

**Table 1 healthcare-12-02134-t001:** Demographic information.

Characteristic	Frequency	Percentage
Age		
21–24 months	9	25.7%
25–36 months	14	40%
37–48 months	4	11.4%
49–60 months	4	11.4%
61–72 months	2	5.7%
73–84 months	1	2.9%
Gender		
Male	21	40%
Female	14	60%
Ethnicity		
White/Caucasian	25	74.3%
Black/African American	3	11.4%
Hispanic/Latino	3	11.4%
Asian/American	1	2.9%
Hemiplegic Side of the Body		
Left	16	45.7%
Right	19	54.3%
MACS/Mini-MACS Level		
1	9	25.7%
2	18	51.4%
3	8	22.9%
Cause of Hemiparesis		
Prenatal CVA	10	28.6%
Perinatal CVA	10	28.6%
Neonatal Ischemia	5	14.3%
Acquired CVA during Infancy	1	2.9%
Neonatal Leukomalacia	2	5.7%
Brain Malformation	1	2.9%
Traumatic Brain Injury (TBI) due to GSW	1	2.9%
Non-accidental TBI	3	8.6%
Infection (MMSA, encephalitis)	2	5.7%
Secondary Diagnoses		
Expressive/Receptive Language Disorder	5	14.3%
Phonological Disorder	2	5.7%
Blood Clotting Disorder	1	2.9%
Hydrocephalus	1	2.9%
Epilepsy	3	8.6%
Prematurity	4	11.4%
Hemispherectomy	2	5.7%
Autism Spectrum Disorder	1	2.9%
Dystonia	1	2.9%

Note. MACS = Manual Ability Classification System; Mini-MACS = Mini-Manual Classification System; CVA = Cerebrovascular Accident; TBI = Traumatic Brain Injury; GSW = Gun-shot wound; MMSA = methicillin-susceptible staphylococcus aureus infection.

**Table 2 healthcare-12-02134-t002:** Sample intensive, group-based PCIMT summarized lesson plans.

Weekly/Daily Theme: Storybook—We’re Going on a Bear Hunt	Weekly Theme/Daily Theme: Disney/Frozen/Frozen II	Weekly Theme/Daily Theme: Oh, the Places You’ll Go! (Final Day)
Focus: CIMT	Focus: CIMT	Focus: Bimanual
Hour 1: OT/PT Warm-up ROM song Reach/point/turn pages of story/perform movements from story Obstacle Course: FM: Reach, grasp, and release of large and small bears GM: stooping, stepping up, crawling over crash pad, and tall kneeling at a bench while carrying	Hour 1: OT/SLP Warm-up ROM song Reach/point/turn pages of story, singing movie songs Stations: 4 Stations involving: dress-up, pulling characters on sled, coloring, Frozen matching game involving reach, grasp, release, pulling, pushing, supinating, and stabilizing actions	Hour 1: OT only Warm-up ROM song Reach/point/turn pages of story Individual short term goal check—1:1 ratio Oh, the Places You’ll Go! Obstacle course: FM: Pulling/Pushing, grasp, release, supinating with resistive toys GM: creeping through tunnel, walking across balance beam, propelling scooter board, carrying
Hour 2: OT only Prepare/eat snack-spreading peanut butter, grasp/release banana slices Clean table, wash hands, diaper change/use restroom	Hour 2: ½ OT/SLP and ½ OT only Make “Frozen playdoh” Eat Frozen popsicles Clean table, wash hands, diaper change/use restroom	Hour 2: OT only Snack-decorate cookies-spreading icing, shaking sprinkles using both hands Clean table, wash hands, diaper change/use restroom
Hour 3: ½ OT Only; ½ PT and adaptive martial arts FM Bear Hunt Craft Adaptive Martial Arts Arm movements-various punches Leg movements-kicks, balancing on alternating legs	Hour 3: OT only 3 FM stations: making Olaf craft, squirting ice cubes across tray with spray bottle, playing “Don’t Break the Ice” Dress up in winter clothes and have a snowball fight (crumpled paper)	Hour 3: OT only Independence Day FM craft-Firecracker shaker tubes (gluing, coloring, peeling stickers, scooping beans, ripping tape) Dance party with kid music and bubbles/water play using both hands
Discuss activities/targeted movements with caregivers at end of session	Discuss activities/targeted movements with caregivers at end of session	Discuss daily activities, provide caregivers a univalve cast, send specific HEP within 2 weeks

Note. PCIMT = Pediatric constraint-induced movement therapy; OT = Occupational Therapy; PT = Physical Therapy; SLP = Speech Language Pathology; ROM = Range of motion; FM = Fine motor; GM = Gross motor; HEP = Home Exercise Program. During handwashing, the non-affected UE turned on the faucet, while the affected UE scrubbed soap using isolated finger movements with assistance.

**Table 3 healthcare-12-02134-t003:** Unilateral function and bimanual coordination means, standard deviations, and mixed Factorial ANOVA results.

Repeated Measures Effects							
Variable	Pre-Intervention		Post-Intervention		*F*(1, 35)	*p*-Value	η^2^
	M	(SD)	M	(SD)			
QUEST DM							
	48.41	(17.03)	57.37	(19.88)			
4–6-year-old	58.31	(16.12)	54.9	(20.1)			
Total	48.38	(16.51)	56.59	(19.68)	7.987	0.008 *	0.195 **
QUEST Grasps							
2–3-year-old	45.11	(19.02)	48.72	(20.13)			
4–6 year-old	43.1	(27.33)	39.73	(22.11)			
Total	44.47	(21.57)	45.89	(20.87)	0.001	0.971	0.000
QUEST WB							
2–3-year-old	48.64	(30.7)	67.31	(21.78)			
4–6-year-old	52.32	(28.73)	54.91	(28.46)			
Total	49.8	(29.72)	63.41	(24.35)	7.613	0.009 *	0.187 **
QUEST PE							
2–3-year-old	30.07	(23.63)	46.46	(22.51)			
4–6 year-old	25.81	(18.66)	36.71	(25.92)			
Total	28.73	(22.01)	43.4	(23.71)	11.606	0.002 *	0.26 **
QUEST Total							
2–3-year-old	44.82	(17)	54.97	(17.18)			
4–6 year-old	42.85	(19.82)	47.11	(19.67)			
Total	44.2	(17.66)	52.5	(18.09)	13.889	<0.001 *	0.296 **
AHA							
2–3-year-old	33.5	(19.45)	47.5	(22.48)			
4–6-year-old	34.36	(19.74)	42.55	(21.24)			
Total	33.77	(19.74)	45.94	(21.91)	67.455	<0.001 *	0.671 **
Interaction Effects							
QUEST WB					4.35	0.045 *	0.117
AHA					4.641	0.039 *	0.123

Note. η^2^ = Partial Eta Squared; N = 35; QUEST = Quality of Upper Extremity Skills Test; DM = Dissociated Movements subtest; 2–3-year-old = 2–3-year-old pCIMT program (n = 24); 4–6-year-old = 4–6-year-old pCIMT program (n = 11); WB = Weightbearing subtest; PE = Protective Extension subtest; AHA = Assisting Hand Assessment; * = Statistically significant difference (*p* < 0.05); ** = large effect size of η^2^ > 0.14.

**Table 4 healthcare-12-02134-t004:** Occupational performance means, standard deviations, and statistically significant mixed factorial ANOVA repeated measures effects.

Variable	Pre-Intervention		Post-Intervention		F(1, 35)	*p*-Value	η^2^
	M	(SD)	M	(SD)			
COPM Performance							
2–3-year-old	15.17	(5.2)	28.29	(7.19)			
4–6-year-old	12.81	(4.9)	26	(6.18)			
Total	14.43	(5.15)	27.57	(6.88)	118.78	<0.001	0.783 *
COPM Satisfaction							
2–3-year-old	15	(7.08)	32.33	(8.21)			
4–6-year-old	13.27	(5.18)	30.64	(7.8)			
Total	14.45	(6.52)	31.8	(8.01)	105.97	<0.001	0.763
PEDI SC Functional Skills Normative							
2–3-year-old	37.92	(10.8)	38.67	(12.04)			
4–6-year-old	16.01	(7.26)	20.46	(9.22)			
Total	31.03	(14.21)	32.95	(14.02)	4.18	0.049	0.112
PEDI SC CA Scaled							
2–3-year-old	47.76	(6.34)	50.36	(7.95)			
4–6-year-old	51.34	(8.79)	51.37	(16.41)			
Total	48.88	(7.26)	50.68	(11.05)	5.023	0.032	0.132 *
PEDI SF Functional Skills Normative							
2–3-year-old	44.6	(13.89)	47.69	(14.88)			
4–6-year-old	24.56	(8.8)	29.79	(11.68)			
Total	44.2	(17.66)	52.5	(18.09)	7.201	0.011	0.179 *

Note. η^2^ = partial eta squared; N = 35; COPM = Canadian Occupational Performance Measure; 2–3-year-old = 2–3-year-old pediatric Constraint Induced Movement Therapy (pCIMT) program (n = 24); 4–6-year-old = 4–6-year-old pCIMT program (n = 11); PEDI = Pediatric Evaluation Disability Inventory; SC = Selfcare; CA = Caregiver Assistance; SF = Social Function; * = large effect size of η^2^ > 0.14. All other subdomains of the PEDI produced non-statistically significant changes.

**Table 5 healthcare-12-02134-t005:** Occupational performance means, standard deviations, and statistically significant mixed factorial ANOVA between-subject effects.

Variable	2–3-Year-Old Program		4–6-Year-Old Program		F(1, 35)	*p*-Value	η^2^
	M	(SE)	M	(SE)			
PEDI SC Functional Skills Normative							
Total	38.29	(2.05)	18.24	(3.02)	30.24	<0.001	0.478 *
PEDI SC CA Normative							
Total	39.45	(2.19)	20.10	(3.23)	24.66	<0.001	0.428 *
PEDI SF Functional Skills Normative							
Total	46.15	(2.57)	27.18	(3.8)	17.10	<0.001	0.341 *
PEDI SF Functional Skills Normative							
Total	47.65	(2.83)	32.44	(4.18)	9.07	0.005	0.216 *

Note. η^2^ = partial eta squared; N = 35; 2–3-year-old = 2–3-year-old pediatric Constraint Induced Movement Therapy (pCIMT) program (n = 24); 4–6-year-old = 4–6-year-old pCIMT program (n = 11); SE = standard error of measurement; PEDI = Pediatric Evaluation Disability Inventory; SC = selfcare; CA = Caregiver Assistance; SF = Social Function; * = large effect size of η^2^ > 0.14. All other PEDI scores revealed non-statistically significant between-group differences.

## Data Availability

The de-identified data set for this study may be obtained per request of the author at katherine.ryan-bloomer@rockhurst.edu.

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
