# Peer review of "Young Children Benefit from Intensive, Group-Based Pediatric Constraint-Induced Movement Therapy"

_healthcare, 2024, doi:10.3390/healthcare12212134_

Round 1
Reviewer 1 Report
Comments and Suggestions for Authors
Comments to the Author
Overall, the manuscript is well-written, and the outcomes are significant due to their scientific and clinical relevance. It is suitable for publication in this journal; however, some definitions and methodological details need clarification. I recommend accepting the manuscript after addressing the following revisions.
1. Intro. P.1, line 40. The placement of the abbreviation "UE" is incorrect.
2. Intro. P.2, line 65. Dosage (or you can say intensity) usually refers to the total number of training hours (e.g., 90 hours). I recommend using "dosage schedule" to refer to the different intervention frequencies and durations in various CIMT models, to avoid confusion. Therefore, this topic sentence should be revised for clarity.
3. Intro. P.2, line 84. Challenges of Signature pCIMT. One significant difference in implementing CIMT between children and adult populations is compliance. Therefore, CIMT for children often needs to be combined with engaging scenarios or themes. You can further elaborate on related issues here.
4. Intro. P.3, line 118. I agreed with “the literature reveals much variation with regards to the intensity of dosage and involvement of bimanual therapy within group-based pCIMT programs.” How did you determine the intensity of dosage in this study?
5. Materials and Methods. P.3, line 137. Do not abbreviate (HP) the first occurrence in the text.
6. Materials and Methods. What were the differences in the lessons between the 2-3-year-old group and the 4-6-year-old group? Or what modifications were made to the intervention for the younger group? How can the “wash hands” activity be implemented in CIMT?
7. Materials and Methods. Line 177. Were the assessors blinded for scoring AHA videos?
8. Materials and Methods. Line 226. This program included small weekend home exercise activities. Did the authors require caregivers to document treatment dialogue for each weekend exercise to ensure protocol fidelity?
9. Materials and Methods. This study is highly valuable in clinical practice, but I am curious whether restricting the less-affected hand of preschool children (especially those around 2 years old) for 24 hours has caused any adverse effects. If so, how did you overcome them? If these aspects were described in more detail, the value of this paper would be even higher.
10. Data analysis. Line 239. “the PI ran between MANOVAs to determine if there were differences in outcome measures based on age and developmental level of the two pCIMT programs” Did the authors use individual 2*2 analyses of variance (ANOVA) for each outcome measure? Please specify.
11. Results. Line 249. Do the overall results include both the MANOVA and the individual ANOVA outcomes? This part needs some further clarification.
12. Results. Line 267. The accepted hypothesis should not be mentioned in the results section. It would be suitable to mention it in the Discussion section.
13. Results. Line 272. What is Univariate MANOVA? Do the authors mean the ANOVA?
14. Results. Line 288. It is incorrect to state that the PEDI has eight subdomains. Additionally, the author mentioned that there are three domains in the PEDI.
15. Discussion. Line 325. I do not understand the term “anecdotal gains”.
16. Discussion. Line 345. The authors stated that “The PI, who was AHA certified, was more keenly aware of the skills to target during the program versus the clinical investigator who ran the 4-6-year-old program, who was less familiar with the AHA and thus may not have targeted the specific AHA bimanual skills as heavily.” Therefore, it would be anticipated that the 2-3-year-old group may have larger improvements than 4-6-year-old group. However, I do not find this pattern in the results. I recommend deleting this statement.
Author Response
Here is a revised version of the manuscript. I sent you my revisions before I realized there were 2 additional reviewers. One of the reviewers requested major revisions to the results section after consulting with statisticians. I have subsequently revamped the manuscript. the manuscript. I wanted to resend you my updated responses since the manuscript has changed significantly. Thank you so much for your consideration of the most recent revisions!

Reviewer 2 Report
Comments and Suggestions for Authors
Dear Authors,
A couple of quick points and then on to the main reason why I cannot adequately review your manuscript.
In the introduction please improve your list of requirements for CIMT. Both a and c focus on dosage. A should be about shaping techniques only.
line 95 the header is formatted incorrectly.
The major problem with the manuscript is that your statistics are in correct or at least your understanding of your statistics are in correct. Therefore your results and interpretation of your results need to be completely re-written.
When you run a MANOVA with SPSS you are correct in the overall description you provide that it creates a model that considers all dependent variables at one time. MANOVA however does NOT ever provide univariate analysis per variable. By definition univariate analyses focus on one dependent variable at a time or are univariate ANOVAs.
Now, SPSS may have provided you with the MANOVA results followed by ANOVA results if you have the correct selections, although, I cannot confirm or deny what procedures you took, given your presentation. There is no such thing as a univariate MANOVA.
You also discuss 'gains', but you did no analyses on change or gain scores. You appear to have completed repeated measures analyses.
While more articles on practice-based evidence on P-CIMT are needed in the literate. I have no way of understanding if your paper could add value in its current version. I suggest you work with a statistician.
Author Response
Thank you for taking the time to review this manuscript and provide feedback. Please see the attached document that provides a point-by-point response to the comments. Thank you!

Reviewer 3 Report
Comments and Suggestions for Authors
Review Report
Summary
Overall, this article has merit. Furthermore, this study aimed to pre-post examine the effectiveness of an intensive, group-based pediatric constraint-induced movement therapy (pCIMT) program for young children. The program ran for 4 weeks with 3 hours of therapy per day, 5 days per week with 3 weeks of 24-hour casting for the unaffected arm, followed by 1 week of bimanual focus.
The key findings, which are of significant importance, indicated that the intensive, group-based pCIMT clinic model was effective and feasible to implement with support from various stakeholders. However, as the authors mentioned, there are several limitations.
Title
The title's clarity is obvious and draws the reader's attention. In other words, the title describes this research by focusing on the main and most novel findings. Maybe the phrase “preschool-aged children” could be included in the title, as the sample was from 21 months to 6 years old.
Also, the authors should add preschool-aged children to the keywords.
Abstract:
The abstract provides a concise overview of the study. It also describes why it was conducted, the methods the authors used, a summary of the findings with p values, and any implications in academic tone.
The introduction
This section is well-referenced. The authors used 37 relevant references and focused on relevant recent research. The references cited adequately reflect the current state of the field. After careful checking, the reference citation was balanced and fair, with limited self-citation and citation of relevant studies.
The authors provided enough information for the journal’s readership to understand the context. This section accurately describes the up-to-date research on the topic and clearly explains why the study was necessary in lines 117 – 129 (problem, purpose, and hypotheses).
Methods
The study design and methodology were appropriate for reaching the aims described above. The authors have included all the necessary information on samples and methods that would allow other researchers to replicate their study. Specifically, they described the inclusion and exclusion criteria (136 – 142p).
In lines 132 – 134, the authors should add relevant references about the design of this study.
In lines 159 – 161, the authors should add the COVID-19 pandemic to the limitations subsection as a situation that influenced their study.
The methodology was described in sufficient detail, and the researchers used validated instruments. Regarding the ethical part, the authors referred to the Rockhurst University IRB committee's approval for the study (2015-15). In addition, the study was retrospectively registered through Clinicaltrials.gov on March 19, 2024 (ID: 204 NCT06330831).
The statistics are sound, and no uncertainties could be reported. So, this study can be allowed for replication.
Results
The results are presented clearly and well, including both text and tables. They support the main conclusions, and the data seem reasonable considering the study design. The authors used two tables (Table 3 and Table 4) to present their work.
Discussion
The discussion answers to the research hypotheses based on the findings revealed. The authors should discuss further in depth their results in the context of published studies.
Finally, the authors clearly explained the study's implications for the field and its potential future applications.
Author Response
Thank you so much for reviewing the manuscript and providing helpful feedback. Please see the attached document that detail the point-by-point responses to the comments. Thank you!

Round 2
Reviewer 1 Report
Comments and Suggestions for Authors
I appreciate the authors for addressing the previous comments and making the suggested changes. Overall, I find the reporting of this study to be much improved, and the statistical results are now clearer compared to earlier versions of the manuscript. I have just two minor issues remaining:
1. Could you clarify the p-values in each table? It appears that these p-values refer to time main effect, which may cause confusion. I recommend including all relevant p-values in each table, such as those for time and group main effects, as well as the interaction effects.
2. Please double-check the references for accuracy. For instance, references 40 and 51 appear to be duplicates. Ensure all references are unique and properly cited throughout the manuscript.
Author Response
Thank you for the round 2 suggestions. Please refer to the attached document to see point-by-point corrections made based on your suggestions. Thank you.

Reviewer 2 Report
Comments and Suggestions for Authors
Dear Author,
Thank you for the changes made to the manuscript unfortunately, there ae still significant problems.
Your response to the previous comment on the introduction is below.
Comment 1: In the introduction please improve your list of requirements for CIMT. Both a and c focus on dosage. A should be about shaping techniques only. [Paste the full reviewer Response 1: Thank you for inquiring about this. The criteria I used for pCIMT actually came from the Ramey, Coker-Bolt and Deluca book, “Handbook of Pediatric Constraint Induced Movement Therapy (CIMT): A guide for occupational therapy and health care providers, clinicians, researchers, and educators,” published by AOTA, p. 20. Ramey and Deluca list these 5 conditions as essential elements of pCIMT. I have referenced where these criteria originated from on page 1 lines 40-44.
Your introduction still states the principles of PCIMT as follows:
a) intense practice of the hemiparetic UE using behavioral techniques of shaping, feedback, and grading;
b) prolonged constraint of the less-affected UE;
c) high dosage of therapy more than standard therapy;
d) provision of therapy in a naturalistic setting; and
e) incorporation of a transfer package to carry over gains into daily life.
The 5 components you list are not the same as those listed on page 20 of the Handbook of Pediatric Constraint Induced Movement Therapy (CIMT): A guide for occupational therapy and health care providers, clinicians, researchers, and educators primarily because the distinction between the use of a specified procedure to guide the therapy process and the intensity of the process are separate things.. The 5 components listed there are as follows:
1. Constraint of the less-impaired UE; Listed as your b
2. Specific use of systematic shaping techniques to refine and progress behavior; Listed as a, but the concept of “intense practice” should be distinguished from the practice of shaping a behavior. This is the point previously made.
3. High-intensity training; Listed as your c, the authors did intend this to be the ‘dosage’ focus
4. naturalistic setting;
5; transfer package
Materials and methods.
Page 3 line 138-142 is awkward “Time with two levels of pre-and post-intervention served as the repeated measures independent variable and age group (either 2-3-year old program or 4-6-year-old program) served as the between variable. This study examined 16 dependent variables by evaluating subtests and total scores of the four outcome measures used to measure unilateral function, bimanual coordination, and occupational performance.” Specifically, time with two-levels is the most awkward portion.
Why did you choose to not use age as a continuous variable? I think I understand that this is because there were two different camps based on age, but that it not the same as using age as a variable. In essence the two camps are your variable and age is simply a definition that distinguishes those camps. Age within each camp is still a continuous variable, I presume you collected date of birth and dates of treatment.
Reports on age in table 1 do not make sense. For example, “25-36 months (2-2years 11 months)” what does 2-2year 11 months mean?
Since camps are being reported across years did any kids go through both camps?
Data analysis:
You state “if there were differences in outcome measures based on age and developmental level” Did you measure developmental level?
“Though the 2-3-year-old and 4-6-year-old pCIMT programs utilized the same les son plans and flowed similarly.” This does not go in data analysis, but how did you document this? In the previous section you state there was modifications based on age. It appears you need to better describe how your interventions were alike and how they differed.
Discussion section
Adverse events should be reported in the results section.
Author Response
Thank you for the round 2 suggestions. Please see the attached document to see point-by-point corrections based on suggestions. Thank you.

Round 3
Reviewer 2 Report
Comments and Suggestions for Authors
Thank you for making the suggested changes.